# Structural Strategies for Supramolecular Hydrogels and Their Applications

**DOI:** 10.3390/polym15061365

**Published:** 2023-03-09

**Authors:** José Antonio Sánchez-Fernández

**Affiliations:** Procesos de Polimerización, Centro de Investigación en Química Aplicada, Blvd. Enrique Reyna No. 140, Saltillo 25294, Mexico; antonio.sanchez@ciqa.edu.mx

**Keywords:** supramolecular hydrogels, hydrogel chemistry, self-healing hydrogels, adhesive hydrogels, conductive hydrogels, clays, 3D structures

## Abstract

Supramolecular structures are of great interest due to their applicability in various scientific and industrial fields. The sensible definition of supramolecular molecules is being set by investigators who, because of the different sensitivities of their methods and observational timescales, may have different views on as to what constitutes these supramolecular structures. Furthermore, diverse polymers have been found to offer unique avenues for multifunctional systems with properties in industrial medicine applications. Aspects of this review provide different conceptual strategies to address the molecular design, properties, and potential applications of self-assembly materials and the use of metal coordination as a feasible and useful strategy for constructing complex supramolecular structures. This review also addresses systems that are based on hydrogel chemistry and the enormous opportunities to design specific structures for applications that demand enormous specificity. According to the current research status on supramolecular hydrogels, the central ideas in the present review are classic topics that, however, are and will be of great importance, especially the hydrogels that have substantial potential applications in drug delivery systems, ophthalmic products, adhesive hydrogels, and electrically conductive hydrogels. The potential interest shown in the technology involving supramolecular hydrogels is clear from what we can retrieve from the Web of Science.

## 1. Introduction

Supramolecular chemistry is considered one of the most important areas related to many classes of materials including hydrogels for use in drug delivery and as biomaterials. This, coupled with the practice of cross-linking based on host–guest recognition, provides supramolecular hydrogels with the underlying affinity and dynamics to control key factors for the stability of their mechanical properties [1,2] as well as offering the ability to protect and control the release of therapeutics [3].

In the construction of a network of supramolecular hydrogels, a gelling molecule or macromolecule is necessary to form non-covalent intermolecular dynamic bonds such as hydrogen bonds, van der Waals interactions, electrostatic interactions, hydrophobic interactions, π−π bonds, metal–ligand coordination, and host–guest interactions [4].

On the contrary, hydrogels built under purely covalent chemical structural considerations present very different properties to those built under supramolecular architectures, and, evidently, they present different morphologies, and their molecular interactions with other molecules can be abysmal. It should not be neglected that any non-covalent interaction is a synergy that gives rise to gelation [5].

There are structural differences between the macromolecular hydrogel and supramolecular hydrogel structures, the former of which, on the one hand, is a cross-linked amorphous structure that can be formed during polymerization using multifunctional monomers that have cross-linking action, and, on the other hand, cross-linking can also be obtained through a chemical reaction on an existing polymer. Eelkema and Pich mention acrylamide polymerized with an *N*,*N*′-methylenebis(acrylamide) cross-linker as a very specific example of a macromolecular hydrogel that presents characteristics of a swelling ratio of 13.3 and an E-modulus of 0.03 MPa. Meanwhile, supramolecular hydrogels are designed with oligomers or polymers that self-assemble in minutes or hours into ordered superstructures through non-covalent interactions that subsequently cross-link through non-covalent interactions, entanglement, or covalent means that then lead to gelation. A typical example is the protonation-induced self-assembly and subsequent gelation of hydrophobic dipeptide derivatives in water, resulting in transparent gels with low yield stress [6].

Multicomponent structural characteristics can be given in supramolecular self-assembly and are ubiquitous in, for instance, synthetic processes leading to the formation of highly ordered and complex architectures. The cooperative and synergistic noncovalent interactions between multiple building blocks in supramolecular assemblies impart a harmonic dynamic of significantly flexible connections, presenting many opportunities for the present and future to take full advantage of supramolecular combinations and interactions in order to develop more specific hydrogels with finely tailored properties. The structurally relevant impacts on the specific functionalities of robust supramolecular hydrogels fall on applications as delicate as biomedicine. Interestingly, some of the most important structural characteristics of supramolecular compounds depend on the raw materials, which have a great impact on their structural chemistry and molecular interactions, all of which also depend on factors such as the concentration of reagents, temperature, pH, and reaction medium, and these conditions of synthesis have a great influence on the thermodynamics and kinetic parameters.

One characteristic of molecular hydrogels lies in the fact that they are materials that have a three-dimensional network structure and can hold or absorb water in each cross-link junction, wherein these unbound water molecules show no evidence of order. The design of the adhesion junctions depends on the way in which the adherends are joined to give rise to a topology and, obviously, to surface properties in relation to adhesion [7].

Hydrogels are widely used in tissue engineering, implantable devices, and drug delivery systems due to their biocompatibility and controllable physical and mechanical properties [8]. Hydrogels include cross-linked three-dimensional (3D) networks containing an extensive range of structural forms and chemical compositions [9,10]. The static behavior of conventional hydrogels and the dynamic behavior of smart hydrogels are two extremes of behavior that can significantly affect the performance of drug delivery systems, among other technologies. In this sense, the dynamic nature of supramolecular chemistry is applicable to the development of smart hydrogels. Indeed, every time new technologies advance for specific demands, the structural chemistry of hydrogels is emphasized [11]. In comparison with supramolecular hydrogels, the specific properties of conventional hydrogels are much less easily achievable as the polymer network is permanently joined together by strong and irreversible covalent bonds [5].

Recently, reviews have been published related to molecular assemblies obtaining supramolecular structures that can be supported on a solid support for the self-assembly of more sophisticated sensors [12]. In other studies, supramolecularity is taken to exploit enzymatic catalysis in polyene cyclizations, achieving the transformation of a variety of alkenes into closed-ring products with high stereocontrolled efficiency [13]. Additionally, the high reactivity of the carboxylic acid functional moiety and of the structural features of non-polar alkane chains are being used to synthesize 2D supramolecular assemblies, and, in this way, the tunable compound 4-(decyloxy)benzoic acid on a surface of Au(111) was designed [14]. Following the same type of investigations, difluorobenzothiadiazole and quaterthiophene (PffBT4T-2DT) thin films were made to study supramolecular assembly on flat Si/SiO_2_ as a substrate. It is worth mentioning that surface roughness significantly hinders self-assembly in the designing of organic semiconductors [15].

This review addresses issues around macromolecular and supramolecular structures that are dominated by nanoscience and even nanotechnology since they allow a straightforward transition between different length scales. The research on biointerfaces comprises the development of specific molecular patterns found in complex biological systems and their applications in the controlled release of therapeutic agents. This discipline of technology integrates molecular assembly and nanoscale design to provide control over biological processes. It is also worthy, as far as possible, to analyze the progress of significant technological advancements in supramolecular materials boosted by hydrogels and their derivatives. This review greatly highlights the value of supramolecular hydrogels and presents a set of rational guidelines for the future development of these remarkable compounds of defined molecular architectures, even beyond tissue engineering applications, as well as their implications for pharmacological and clinical effects in future research.

In sum, the topics addressed in this review manuscript are described below, paying close attention to the current developments in supramolecular hydrogel technology, which are supported by carefully selected. Furthermore, it is essential to know that the potential interest aroused by the technology involving supramolecular hydrogels is clear from what we can retrieve from the Web of Science [16].

The manuscript begins with an explanation of the relationship between hydrogels and drug delivery systems. Subsequently, the 3D structures and their implicit factors are described. As a subsequent point, the chemical effects of the use of clays in hydrogels are specified. Likewise, selected applications for supramolecular hydrogels are described. At the end of the manuscript, future perspectives on the exciting field of supramolecular structures and their relationship with hydrogels are detailed.

## 2. Hydrogels and Their Relationship with Drug Delivery Systems (DDSs)

The field of tissue engineering comprises a constant evolution that involves the regeneration of functional tissues [17]. Drug delivery systems are intended to deliver a proper dosage during a given time interval and to have the spatial control to deliver pharmaceutical agents to precise sites in the human body. This spatiotemporal control of any drug release can reduce the off-target release and subsequent degradation of active therapeutic agents upon administration by protecting the payload from harsh in vitro or in vivo environments. Additionally, it is worth noting that the inclusion of a photosensitive element allows for the delivery of higher concentrations of a drug to the target site in a more controlled manner. This peculiarity of drug manipulation enhances its effectiveness and, at the same time, minimizes off-target toxicity [18]. In an extensive review of Bernhard et al., two main areas of research on drug delivery systems are specified: the use of supramolecular binding systems to design hydrogels as injectable materials for local drug delivery, and the adaptation of supramolecular hydrogels in drug delivery by adjusting the affinity between the drug and the material [19].

These characteristics impact and particularize the chemical properties that can dictate the way in which hydrogels can be delivered into the human body. It is such that hydrogel delivery systems can be classified based on their size as macroscopic hydrogels, microgels, and nanogels [20]. However, regardless of the variety of packaging, transport, and storage, both the drug and the hydrogel must be chemically and physically stable. Hydrogels can be delivered via several means, for instance, surgical implantation, local needle injection, or systemic delivery via intravenous infusion.

It is such that several enhanced biomaterial systems with controlled release properties guarantee the efficacy of therapies. At the same time, it is plausible that a broad range of possibilities for drug carrier interface modification always keeps side effects under control and at very low levels [21]. A miscellaneous array of nanostructures for drug loading that are inherent to the incorporation of stimuli-responsive agents are illustrated in Figure 1. In a study directed by Vermonden, an overview of supramolecular hydrogels containing hyaluronic acid (HA) as one of the building blocks is specified. The authors argue that HA being present in many tissues and the extracellular matrix (ECM) makes it amenable to the designing of supramolecular materials for biomedical applications and that having excellent processing properties makes these more attractive [22]. Furthermore, this approach is mentioned in a very descriptive work in which it is emphasized that to enhance the mechanical properties of hydrogels prepared through self-assembly, it is necessary to increase the ability to fabricate complex and precise hydrogels tailored to drug delivery applications [23].

In order to structure an injectable hydrogel platform based on hydrophilic thermosensitive linear–bottlebrush–linear (LBL) triblock copolymers that self-assemble at body temperature, a research group developed systems to systematically control the mechanical mismatch in an implant in a controlled water environment and other factors such as low viscosity at high concentrations of a hydrogel to facilitate injectability, physical cross-linking to eliminate any percolation of a chemical into the body, and mechanical resilience [24]. In addition, in 2020, Ahsan et al. reported that a thermosensitive, physically cross-linked injectable hydrogel was formulated for the effective and sustained delivery of disulfiram (DSF) to cancer cells [25]. With the same idea, a polyhistidine (Phis)-based Ni-coordinated hydrogel with physiologically pH responsiveness was prepared [26]. Alternatively, a series of injectable conductive self-healing hydrogels have been investigated based on quaternized chitosan-g-polyaniline (QCSP) and poly(ethylene glycol)-co poly(glycerol sebacate) (PEGS-FA) for cutaneous wound healing [27]. The adequate incorporation of chemically cross-linked domains and physically cross-linked domains effectively enhances the mechanical properties of double-cross-linked (DC) cellulose hydrogels [28]. In the same way, as the capabilities of hydrogels have dramatically increased over time, they have, unsurprisingly, become useful tools for a wide range of fields and disciplines [29,30]. This is why studies by researchers such as Correa et al. primarily focus on the contributions of injectable hydrogel systems, with an admirable emphasis on dynamic hydrogels [31].

The combined nanoparticle–hydrogel structures typically used in conventional drug delivery can modulate the release kinetics of immunotherapeutic agents and function as tissue scaffolds [32]. Based on this, some drugs interact with enzymes in ways that change the ways the enzymes perform their jobs. In this category, a particular emphasis on discussing the advantages of the way in which a compound could interact with an enzyme in a way that increases its expression, known as induction, or in a way that reduces or blocks its function, referred to as inhibition, is of valuable importance. This means that one drug’s impact on an enzyme or transporter could alter another drug’s pharmacokinetics. In these scenarios, the former is commonly known as the perpetrator drug, which negatively affects the processing and physiological effects of the second drug. For the above scenario, some research that involved chitosan/glycerophosphate (CS/GP) hydrogel/microparticle assembly loaded with lornoxicam, a non-steroidal anti-inflammatory drug, presented a lower concentration of it into the blood, not via a direct injection of a lornoxicam solution into the delivery site, which is considered proof of the sustained delivery of this drug. In this setting, microparticles were embedded into the hydrogel before its gelation process by mixing the microparticle suspension and the CS solution. This method was used with thermosensitive hydrogels, for which the microparticle suspensions were pre-formed, controlling the microparticle size to later mix them with the CS solution [33].

Other authors’ strategy was to induce chemical and physical cross-linking domains within chitin hydrogels via sequential chemical cross-linking with epichlorohydrin (ECH), involving hydrogen bonding, hydrophobic interactions, and the formation of crystalline hydrates in an aqueous ethanol solution. Due to behavior coordinated to avoid stress concentration, there was an increase in the stress at fracture, higher deformation, and an energy dissipation mechanism [34].

Physical networks typically employ enthalpy-dominated cross-linking interactions. Therefore, through the mathematical relationship between the cross-linking interaction thermodynamics and viscoelasticity of the resulting physical networks, an entropy-driven physical network based on dynamic and multivalent polymer–nanoparticle (PNP) interactions were created for hydrogels that exhibit temperature-independent viscoelasticity [35]. Zeng et al. explored His-Zn^2+^ binding and showed that the coordination number is an important parameter for the mechanical strength of a hydrogel that could achieve mechanical features including a Young’s modulus of 7–123 kPa, fracture strain of 434–781%, and toughness of 630–1350 kJm^−3^ [36].

In 2018, Caruso and co-workers implemented a supramolecular gel medium for the crystallization of active pharmaceutical ingredients (APIs) to control crystal size, morphology, and polymorphism, as these features determine the performance of pharmaceutical formulations. They reported the crystallization of APIs in a titanium (IV) (Ti^IV^) medium with a natural polyphenol-like tannic acid, without the need for a heat–cool trigger, which was used as a medium for API crystallization [37].

Personalized therapies are extremely diverse, which opens opportunities in a wide range of options for a diversity of drugs, including drug toxicities, pharmacokinetics, and biodistribution. In other instances, material-based strategies, such as nanoparticles derivatized with targeting ligands, are less specific than antibody–drug conjugates and, consequently, could be best used to deliver cargo with more specificity to molecular or cellular targets [38]. In this context, the release of a drug with high specificity must be constant and pharmacokinetically controlled. It is in this way, and with the advent and discovery of new drugs to attack complex diseases, that perfectly designed materials must be outlined. The required duration of a drug’s availability and its release profile, continuous or pulsatile, depend on the specific application. As a drug decreases at the indicated site, the hydrogel should be structured either to degrade to avoid surgical removal or to be reused via drug refilling or adapted for tissue regeneration.

## 3. D-Printed Hydrogels

Three-dimensional structures are the product of a series of two-dimensional (2D) layers. This approach, therefore, has obvious advantages, and, to tell the truth, the printing process involves simultaneous deposition that entails, for instance, cells and biomaterials, using systems of high precision to build architectures in a layer-by-layer fashion [39,40]. Using conventional techniques, it is practically impossible to structure highly aligned supramolecular domains to reproduce 3D model hydrogels. In order to reduce non-reproducibility, Sather et al. designed a method for the 3D printing of ionically cross-linked liquid crystalline hydrogels from aqueous supramolecular polymer inks, wherein the pH is important to establish intermolecular interactions between the self-assembled structures [41]. At the same time, a strategy for the specific hydrogels is currently being used in 3D printing scaffolds that may be employed to create scaffolds effective of stimulating the alignment of cells and fabricate materials with anisotropic ionic and electronic conductivity [40].

The designing of 3D-printed living materials is led by quantitative models that establish the responses of programmed cells in the printed microstructures of hydrogels [42,43]. With the non-submerged printing approach, a highly viscous alginate and methylcellulose (Alg/MC) hydrogel blend and a cell–GelMA mixture were printed alternately on top of each other in order to obtain cell-laden structures. The gelatin methacrylate (GelMA) reinforced the whole structure after printing, and, additionally, the cross-linking was enhanced using UV light [44]. A feasible “gelation and soaking” method was applied to convert chitosan–gelatin (CS-Gel) composite hydrogels to stiff and tough chitosan–gelatin–phytate (CS-Gel-P) hydrogels in a sodium phytate solution [34]. Considering the further expansion of the scope in the responses of hydrogels according to their natural chemical properties, the Sun research group has successfully been exploring the tetramerization of adenosylcobalamin (AdoB12)-dependent C-terminal adenosylcobalamin-binding domain (CarHC) proteins, which is essential for the formation of an elastic hydrogel that can undergo a rapid gel–sol transition caused by light-induced disassembly [45].

In a recent contribution to item 3D bioprinting, a system composed of oxidized hyaluronate, glycol chitosan, adipic acid dihydrazide, and alginate (OHA/GCS/ADH/ALG) hydrogels for potential applications in cartilage regeneration was investigated in vitro. The study emphasizes that using ALG and calcium ions enhanced the mechanical stiffness of the resulting material [46]. In addition, significant studies based on the viscosities and shear moduli of alginate and alginate/carrageenan hydrogels with different printing parameters were evaluated [47].

As progress has been made in the design of matrices for 3D cell cultures, strategies have emerged such as cell spreading, migration, signaling, and mechanotransduction under physiological conditions. The flexibility of many “smart” materials has allowed the integration of responsive moieties into specific biological targets, particularly for drug delivery applications [48], in order to achieve specific interactions and responses from cellular systems by regulating key variables of chemokines and cytokines [49] because, in biological materials, cross-links are often considerably weaker than the covalent backbones that hold the structures together [50].

A 3D printing of self-healing hydrogels obtained by vat photopolymerization (VP) process has been developed at room temperature without any stimulus. Excellent results were obtained through dispersive forces using poly(vinyl alcohol) (PVA) and photocurable species, such as acrylic acid (AAc) and poly(ethylene glycol) diacrylate (PEGDA) [51].

An outstanding study describes design considerations for gel-phase materials, such as self-healing and slimming bioinks, with a focus on their dynamic, biochemical, and mechanical gel properties. These inks take advantage of the ability of guest–host complexes to disassemble upon the application of physical force and then to reform once the force is removed [52].

Following the basic design principle of 3D printing, which is to digitally cut a 3D object into multiple thin 2D layers and then recreate that object by depositing materials in a layer-by-layer 2D pattern, suitable structures have been designed for the immobilization of enzymes in a complete framework (post-impression) or incorporated into impression materials during framework fabrication (entrapment) [53].

Four-dimensional bioprinting, using smart materials that can be stimuli-responsive, could also be used to create dynamic 3D-printed biological architectures adapted by changing their sizes and shapes based on an applied stimulus [40]. In addition, photo-cross-linkable hydrogels and natural proteins are among the most well-known materials for the fabrication of microrobots using 3D printing through a photochemical reaction method, with multiple applications, from environmental to biomedical [54] ones, as can be seen in Figure 2.

Of note is the development of a partially automated portable bioprinter capable of in situ printing and cross-linking hydrogel scaffolds. The printer includes an ultraviolet light source to photocross the bioink hydrogels deposited on the geometry of surfaces that could present some defect. Surfaces may not necessarily be flat at the site of an injury [55]. Several factors are driving profound changes in the way chemical scientists organize their workflow, whether in the basic researching of hydrogels or in obtaining supramolecular hydrogels for specific applications. To be explicit, this requires increasingly sophisticated molecular construction to self-assemble diverse molecules with diverse properties.

## 4. Clays into Hydrogels

Successful formation of nanocomposite particles has recently been evidenced in some works on Laponite^®^. The studies of Boyes and Thorpe found that the combined effects of hypoxia [5% O_2_] and the structural environment of the hydrogel poly(N-isopropylacrylamide)-*N*,*N*′-dimethylacrylamide-Laponite^®^ (pNIPAM-DMAc-Laponite^®^) could differentiate human mesenchymal stem cells (hMSCs) towards a central gelatinous nucleus pulposus cell phenotype without the need for additional chondrogenic-inducing factors. The inherent simplicity showed that the clinical convenience of such a method could provide an effective and minimally invasive treatment platform for the regeneration of the nucleus pulposus as a treatment strategy for intervertebral disc (IVD) degeneration [56,57]. In an extensive work, Chakraborty et al. reported the surface coating of MOF nanoparticles with Laponite^®^. They chose a ZIF-8 framework (Zn[MeIm]_2_, where MeIm = 2-methylimidazolate). The charge-assisted self-assembly of ZIF-8 nanoparticles with a surface positive charge using Laponite^®^ with negatively charged faces results in a ZIF-8-LP hydrogel nanocomposite. A hydrogel composite of 5-fluorouracil (FU)-encapsulated ZIF-8 (FU@ZIF-8+Laponite^®^) shows the controlled release of FU [58]. In another initiative, Hu’s research group reported a injectable hydrogel sealant with self-contractile characteristic formed via a Schiff reaction between catechol-functionalized CS (CCS) and dibenzaldehyde-terminated polyethylene glycol [59]. The authors found that the mussel-inspired catechol groups and remnant aldehyde groups contributed to the superior wet adhesion to tissues.

On the other hand, Chen et al. propose a postsynthetic modification of the molecular architecture of nanoMOFs using phosphate-functionalized methoxy polyethylene glycol (mPEG−PO_3_) groups to guide the formation of re-dispersible solid materials. The authors mention that these resulting nanoMOFs can favor the loading of drugs [60]. In another contribution, smart hydrogels were fabricated using poly(N-isopropylacrylamide-co-acrylamide) (poly(NIPAM-co-AM)) hydrogels with in-homogeneous structures by assembling subunits of a nano-clay (Laponite^®^) via rearranged strong hydrogen bonding between the polymers and the clay nanosheets of Laponite^®^. The assembled hydrogel complexes can realize a variety of diversified deformations in response to environmental stimuli for use in soft robots, actuators, and artificial muscle [61] (in Figure 3, a systematic explanation of the preparation of a hydrogel is given). In the same circumstances, the ability of the cell morphology, extracellular matrix (ECM) synthesis, expression of chondrogenic marker proteins, and mechanical properties of a newly formed cartilage tissue derived from chondrocytes/hydrogel constructs have been examined [62].

Moreover, there was an increase in the surface area, and the porosity and the clay bonding sites increased their surface area with the mixture of a nano-clay by incorporating it into a hydrogel. Of course, uniformity is a factor that should not be neglected, together with the synthetic way to obtain a clay and hydrogel nanocomposite, for example, via free-radical polymerization or supramolecular assembly [63].

An example of the modulation of the adhesion of hydrogels by adjusting the content of dopamine (DA) and a nano-clay was studied by Gu and Wen and their group of researchers, who specify that there is a dissipation layer by providing the intrinsic strength of the hydrogel, enhanced via the physical cross-linking of Gel and the chemical cross-linking of Gel with polydopamine via a Michael addition or Schiff base reaction and the restriction of the molecular chain induced by the nano-clay [64]. On the contrary, hydrogels made with carrageenan and CS were prepared via electrostatic complexation in aqueous solutions, avoiding the use of chemical reactions or toxic solvents. Additionally, the Lap nano-clay was incorporated as an inorganic material with the potential for drug delivery applications. When the two polysaccharides are both fully charged (pH 5), their electrostatic attraction is very strong, the swelling ratio is lower, and the viscoelastic moduli present a maximum [65]. A supramolecular hybrid hydrogel composed of cucurbit [6], uril, and clay nanosheets contains the electrostatic interactions of a custom-designed cationic copolymer. This hydrogel shows a high mechanical strength (>50 kPa), self-heals rapidly in ~1 min, and dissolves quickly (4–6 min) using an amantadine hydrochloride solution that breaks the supramolecular interactions [66].

On the other hand, the printing of a 3D high-strength supramolecular polymer/clay nanocomposite hydrogel scaffold was proposed. The composition of the hydrogel involving a hydrogen-bonding monomer (*N*-acryloyl glycinamide) (NAGA) and Laponite cross-linkage would eventually result in high-strength and swelling-stable hydrogels providing reliable loading support [67]. There is no doubt that hybrid nanoparticles will provide a pathway to build engineering scaffolds for precise and individualized repairs such as of bone defects and degeneration.

## 5. Selected Applications of Hydrogels Based on Supramolecular Strategies

The supramolecular interaction between some materials and guest molecules has endowed gels with the capacity for reversible gel−sol transformation. A plausible example has to do with self-healing, which is one of the most intriguing characteristics of biological or artificial systems, such as the particular and specific applications described below, due to its intrinsic importance. Thermodynamic stability plays an important role, for example, in the preparation of ophthalmic hydrogels and metallo-supramolecular hydrogels. The above is descriptive of the excellent self-healing ability of various hydrogel systems for an explosive network of applications [68].

### 5.1. Ophthalmic Hydrogels

The first choice for ocular treatment is eye drops, which is the most preferred noninvasive way to deliver drugs for treating ophthalmic diseases. Consequently, it is important for drug transport to overcome multiple ocular barriers, such as the corneal epithelium, choroid, retinal pigmented epithelium, choroidal and conjunctival blood flow, lacrimation, and lymphatic drainage, restricting drug transport into the ocular tissues. Less than 5% of drugs have been reported to enter the aqueous humor, which results in sub-therapeutic drug concentrations in these ocular tissues [69]. To overcome this situation, as a prominent example, polypseudorotaxane hydrogels were developed by mixing polyvinyl caprolactam-polyvinyl acetate-polyethylene (Soluplus^®^) micelles of approximately 99.4 nm with a cyclodextrins solution [70].

In regard to ophthalmic gels, eye drops exist as viscous solutions before application to the eye and are normally used for dry eyes. In situ gels, by comparison, are liquids that are applied as drops to the eye, and it is only after administration that they experience a transition from sol-gel to gel in the conjunctiva through stimuli, such as pH, temperature, or ions, with a significant improvement in the ocular bioavailability [71]. Alginate-based nanoparticles were prepared using W/O emulsion technology and coated with CS and Gel. The mean diameter of the prepared nanoparticles was in the range of 150–270 nm, which is a suitable size for ocular drug delivery that does not cause irritation [72].

Dana, Annabi, and co-workers designed a biocompatible adhesive hydrogel for corneal tissue repair, which can be used for the quick and long-term repair of corneal stromal defects. The bioadhesive hydrogel is made of a chemically modified form of gelatin (Gel) and photoinitiators, which can be photocross-linked after short-time exposure to visible light (450 to 550 nm). Gel was chemically functionalized with methacryloyl groups to form a bioadhesive gel for corneal regeneration (GelCORE). Additionally, the authors show that the physical properties and adhesion strength of GelCORE bioadhesives can be tuned by changing the GelCORE concentration and light exposure time [73]. In this same situation, an efficient antibiotic gel for the clinical treatment of bacterial keratitis was suggested. For such purposes, a supramolecular hydrogel was fabricated via the electrostatic interactions between guanosine 5′-monophosphate disodium salt (GMP) and tobramycin (TOB), which is a kind of aminoglycoside antibiotic bearing multiple primary amine groups in its structure. Obviously, the electrostatic interaction between the primary amine groups of TOB and the negatively charged phosphates on the assembled GMP nanofibers could facilitate the formation of a supramolecular GMP-TOB gel [74], as is well illustrated in Figure 4. Cyclodextrin (CD)-based supramolecular hydrogels were studied for their potential to prolong precorneal retention and increase ocular bioavailability [75]. This effort also opened a beneficial channel for the thermodynamics of an equilibrium system to achieve the gradual and controlled release and capture of cyclodextrin (guest) using a coordination polymer (Mg-CP) as the host and temperature as the stimulus [76].

An excellent supramolecular hydrogel based on β-CD and adamantane for corneal wound healing was well-designed by Fernandes-Cunha et al. In that study, the supramolecular hyaluronic acid hydrogel was formed based on the host–guest interaction. Compared with the free hyaluronic acid solution, the supramolecular hyaluronic acid hydrogel had a prolonged residence time on the cornea without complication [77]. With similar points of view but with different materials, another design architecture included DNA-based supramolecular hydrogels that could form a gel under physiological conditions and provide a promising candidate for a vitreous body substitute because it is colorless and transparent, with a similar density to that of the natural vitreous body [78].

### 5.2. Adhesive Hydrogels

A cooperative structural effect of hydrogels for adhesive applications falls on creating systems for surgical adhesion, tissue engineering, transdermal drug delivery systems, and soft devices such as sensors and actuators, and in the designing of inks for additive manufacturing [79,80,81,82,83,84].

With a high potential impact, significant progress has been achieved in the field of adhesive hydrogels, considerably around supramolecular adhesive hydrogels with inherent applications in tissue engineering. In this regard, an exhaustive review of structural strategies for the formation of supramolecular adhesive hydrogels and their application in tissue engineering was reported by Zhao et al. [85]. Shin et al. demonstrated the feasibility of bioinspired hyaluronic acid modified with a catechol (HA-CA) hydrogel for use in tissue engineering. Furthermore, they showed the adhesiveness of the HA-CA hydrogel [86]. On the other hand, the outstanding gelation of HA-CA and chitosan modified with catechol (CS-CA) allows highly cross-linked systems and an enhanced adhesive ability to tissues to be obtained [87].

Due to their dynamism regarding advanced functionality, a key advantage of supplanting stitches with an adhesive is that an adhesive covers an entire wound as a continuum, because conventional suture materials, such as stitches, staples, or wires, are not the ideal choices to achieve minimally invasive surgeries. The other advantages of gluing the tissue are that the method is quick, saves surgery time, is inexpensive, does not require stitch removal, and is waterproof [88]. In this terrain, the designed adhesive should be biocompatible and nonimmunogenic, should comply with the tissue’s mechanical properties, should develop adhesion over specified time periods, and should adhere strongly, and, in some cases, the adherence must behave reversibly. The designing of hydrogel adhesives is complicated by the inherent nature of these materials, as most of their volume is water and, under physiological conditions, the functional groups needed must have permanent adherence and the covalent junctions be strong and durable; however, they are essentially static and not reversible [7,79], as shown in Figure 5.

In biomedicine, there is specific interest in adhering hydrogels to tissues for the closure of wounds, the sealing of damaged sections of organs [89], and the development of stretchable electronics [90,91]. In addition, a hydrogels’ adhesion to hard and dry surfaces is useful for engineering adaptive and responsive devices. By virtue of this, tissue adhesives require interdisciplinary efforts that span chemistry, mechanics, and biology, as the performance of an adhesive is primarily determined by the adhesive’s physicochemical properties, chemical and mechanical interactions with the tissue, the host immune response, and local environment characteristics [92], and, of course, they are often not properly developed for specific tissue applications [93]. Moreover, it is important to outline that some mechanical relationships do not apply directly to most biological tissues [94]. In contrast, antiadhesion approaches, including the use of impermeable barriers that block fibroblast penetration from surrounding tissues, for preventing postoperative adhesion are, finally, essential, principally, to establish a permanent membrane that can be governed by expanded polytetrafluoroethylene (ePTFE) providing a barrier beneath the sternum, reducing adhesions between the sternum and the epicardium [95]. A hydrogel has been newly developed for uterine applications and to prevent adhesion, and the compound was developed using 3D printing technology to encapsulate human amnion mesenchymal stem cells (hAMSCs) from a human amniotic membrane. In this setting, using natural biopolymers such as Gel and collagen (Col) due to the importance of their unique biocompatibility and biodegradability properties, methacrylated gelatin (GelMA) and methacrylated collagen (ColMA) polymers were synthesized [96]. In addition, Yang et al. studied the effects of imidazolidinyl urea (IU) content and the molecular weight of PEG on mechanical properties. The energy dissipation efficiency and shape memory behavior were also investigated. Moreover, they tested the hemolytic activity, cytocompatibility, in vivo retention time, and tissue compatibility of supramolecular hydrogels [97].

Several studies, such as those reported by Zhang, X. and Zhang, J. and co-workers, detail the reversible adhesive properties of multistimuli-responsive supramolecular hydrogels based on hydrogen bonding. They reported networks formed by adding poly(ethylene polyamine) (PPA) to an aqueous dispersion of oxidized multiwalled carbon nanotubes (ox-MWCNTs). The resulting materials were cross-linked through a combination of weak (N−H···N) and strong (N−H···O) hydrogen bonding [98]. Other findings provided by Burattini et al. include the designing of healable supramolecular polymers from chain-folding polyimide and pyrenyl-functionalized polyurethanes. These materials are held together by aromatic π–π stacking between the π–electron-deficient diimide groups and the π–electron-rich pyrenyl units. It is noteworthy that the π–π interactions are complemented by the hydrogen bonding between the urea and diimide residues [99].

A comprehensive study directed by Liu and his group was focused on using hydroxypropyl-modified α-cyclodextrin (Hy-α-CD) and acrylamide–PEG_20000_–acrylamide (ACA-PEG_20000_-ACA) to construct a polypseudorotaxane with good water solubility. The attractiveness of this research was that through the photo-initiated polymerization of polypseudorotaxane with acrylamide in situ, a capped polyrotaxane cross-linked with 1,4-butanediol diglycidyl ether in a basic solution was obtained to form a slide-ring supramolecular hydrogel that can be stretched to 25.4 times its original length, which recovers rapidly upon unloading. Furthermore, these Ca^2+^-doped hydrogels are used to prepare wearable strain sensors for monitoring human motion [100]. In another line of research, peptide-DNA-based hydrogels that are organized into interlaced superstructures that deconstruct upon the addition of molecules or changes in charge density have been well reported. In this regard, experimental simulations suggest that chemically reversible structures can only occur within a limited range of supramolecular cohesive energies [101].

### 5.3. Self-Healing Hydrogels Systems

Self-repairing behavior originating from dynamic covalent bonds can include disulfide exchange, diselenide exchange, and Schiff bases (imine bonds). Consequently, breaking and remaking covalent bonds require appropriate conditions of reaction. Conversely, in dynamic non-covalent hydrogels, mostly known as autonomous self-repairing hydrogels, the mechanism is based on hydrogen bond interactions, metal–ligand interactions, host–guest interactions, hydrophobic interactions, supramolecular interactions, and the entanglement of polymeric chains, as illustrated in Figure 6 [102].

The specificity of hydrogels for self-repair is due to their reversible properties after being subjected to stress and, evidently, in the nature of their physical bonds. This is how a hydrogel structured using hyaluronic acid behaves together with gallol when exhibiting the spontaneous loading of approximately 93% of a solution of proteins with a concentration of 270 ug/mL in phosphate-buffered saline (PBS, pH 7.4) [103]. In this regard, concerns are reported by Hardman and coworkers, particularly about hydrogels for soft-sensing applications. They explain very accurately the electrical and mechanical properties of an ionically sensorized self-healing Gel hydrogel that at room temperature can undergo strains of up to 454% and, furthermore, presents stability over long periods of time and formidable biocompatibility and biodegradability. The hydrogel is Gel- and glycerol-based and can be 3D-printed to create customizable sensor networks [104].

New research based on supramolecular chemistry establishes that ionic gel self-healing does not require any external stimuli. For instance, polymer design and architecture have a great impact on the creation of ion gels with self-healing abilities as well as other functionalities, such as high stretchability and toughness and compatibility with ionic liquids; herein, the interactions between synthetic polymers such as polyacrylamides and polymethacrylates and an ionic liquid are relevant [105]. In addition, self-healing sodium–cellulose ionic conducting hydrogels (Na-CICH) with an ionic conductivity of ≈ 10^−4^–10^−3^ S cm^−1^ have been developed using an aqueous NaOH/urea cellulose solution system [106].

In another sense, which is no less relevant, the formation of Cu(II) metallohydrogel compounds with third-order nonlinear optical activity based on the self-repairing protein bovine serum albumin (BSA) is pursued in order to produce semiconductor devices due to the large contribution of π-electrons to photosensitivity [107].

Coordination bonds have utility for constructing self-healing polymers and, evidently, because of the presence of metal ligands and metal–ligand dynamic bonds, self-healing polymers can exhibit various functions such as dielectric materials, luminescence, magnetism, catalysis, responsiveness to stimuli, and shape-memory behavior. However, there are coordination compounds that are labile and can undergo a variety of reactions, including electron transfer, ligand exchange, and associated processes. All of these depend on the d-electron configuration of the central metal ion [108,109]. In the event of mechanical damage, chain deformation, or chemical degradation suffered by self-healing hydrogels made up of reversible dynamic covalent or non-covalent bonds, they can easily recover their original properties [102]. This process is the product of the division and reformation of dynamic links in response to various external triggers or initiated autonomously.

The characteristics of a self-healing hydrogel based on the specific coordination of Ni^2+^ with a polymeric ligand made via the polyaddition of 2,6-diethynylpyridine and diazido-poly(ethylene glycol) have been developed. The choice of Ni^+2^ is due to the great selectivity due to the high chain length of the polymeric linker, which can hinder the formation of efficient intermolecular cross-links in Ni^+2^ and 2,6-bis(1,2,3-triazol-4-yl)-pyridine (Ni^+2^-btp) complexes with high thermodynamic stability. The impact on hydrogelation is selective due to metal–ligand coordination interactions between the Ni^2+^ and btp residues, while the additions of other divalent metal ions, such as Zn^2+^, Fe^2+^, Cu^2+^, Mg^2+^, Mn^2+^, Ca^2+^, and Cd^2+^, only induce color changes or precipitation [110].

### 5.4. Electrically Conductiveg Hydrogels

Conductive hydrogels are being used in different scientific settings. Indeed, versatile biochemical sensors using conventional hydrogels based on carbohydrates, polymers, DNA, and peptides are being expansively studied. An illustrative example of the use of a 3D hydrogel based on poly(ethylene glycol) diacrylate through the use of the amyloid beta 42 (Aβ42) peptide for prostate-specific antigen (PSA) detection was reported. In this way, an interdigitated microelectrode (IME) with a dynamic range was substantially improved with an enviable sensitivity for biosensing and has been structured for peptides such as Aβ42 [111].

All cellular communication occurs through electrical signals, impacting critical mechanisms and the functionality of biological tissues. This principle was addressed in skeletal muscle tissue engineering (SMTE) to fabricate in vitro bioartificial muscle tissue to assist and accelerate the regeneration process using electrically conductive hydrogels as the established engineered platforms, which can guarantee a tissue-like microenvironment [112].

Conductive hydrogels can also be obtained by integrating electrical fillers into the hydrogel network. As described by Farr et al., by including graphene oxide (GO) and reduced GO (rGO), metal nanowires, and carbon nanotubes (CNTs) as alternatives, they were able to promote skeletal muscle repair and regeneration [113].

There are three reviews with extensive information regarding the performance of electrically conductive hydrogels in technological applications such as biosensing, flexible electronics, and tissue engineering [114,115,116]. It is noteworthy that the focus of the strategies for producing conductive hydrogels includes the polymerization of a conducting monomer in a prefabricated hydrogel, the mixture of conductive systems via, for instance, the simultaneous or stepwise cross-linking of monomers to synthesize conductive hydrogels, and the self-assembling of modified electrically conductive materials [115]. Figure 7 illustrates the main approaches employed, in another instance, to design electrically conductive hydrogels [116].

In the same direction, electrically conductive hydrogels have been designed for active electrode materials for supercapacitors and lithium-ion batteries with specific morphologies and compositions. Some classic examples of electrically conductive hydrogels are encompassed in Figure 8 [116].

Great developments in conductive nanocomposites have come true, especially those based on CNTs, graphene, transition metal carbides (MXenes), and metal nanofibers. Of these, carbonitrides, nitrides, and transition metal carbides are the most attractive 2D materials, which, together, are known as MXenes. There are various synthetic routes to MXene derived from its precursor MAX phase: M represents a transition metal; A is the main group element, generally, Si or Al; and X is N or C. Through MXene, some ordered-oriented intelligent tunable hydrogel materials (PMZn) with biocompatible polymers and ZnSO_4_ solution as the precursor have been produced. This has led to the manufacture of PMZn-GL conductive hydrogels that can be used as wearable flexible sensors for detecting a series of human activities, such as hand and facial movements [117]. Similarly, the conductive material Ti_3_AlC_2_ precursor is processed into MXene nanosheets that are mixed at the same time with a solution of PVA and ZnSO_4_. The anisotropic PMZn-GL hydrogel obtained presents good tensile properties and conductive faculties and could well be used as a wearable flexible sensor for comprehensive human motion biomonitoring [118]. There is no doubt that the transistor technology used in bioelectronic interfaces, as well as its high spatial resolution in bioelectronic signal recording, is expanding its unprecedented coverage degree [119].

A way to make a patch a biomarker for sweat is to incorporate a transducing layer of nanoporous carbon and MXene (NPC@MXene). The composition of the resulting product is composed of a glucose biosensor along with pH and temperature sensors to precisely quantify glucose that, together with biopotential electrodes, is feasible for recording electrophysiological signals in real-time [120].

Recently, Wu et al. exposed new developments in hydrogel-based triboelectric nanogenerators (H-TENGs). They explain in a pertinent review the recent progress on the subject of flexible TENGs. The authors emphasize the advanced functions to enhance the outputs and stability of H-TENGs in practical applications, such as biomedical electronics, as represented in Figure 9 [121].

In a very eloquent work, 1D-fiber-shaped supercapacitor (FSC) structures were designed with cellulose, composed to produce recyclable ionic hydrogels to establish the foundations of a system of a fully sustainable energy storage fibers made from reused/recycled materials for a wearable Internet of Things (IoT). Such hydrogels were produced with aqueous system cellulose and lithium hydroxide (LiOH) in the presence of urea, and, later, a regeneration process was carried out directly in carbon fiber wire electrodes. In this way, a pair of stretch-broken carbon fiber yarns (SBCFYs) were obtained as current collectors to fabricate 1D FSCs. Additionally, a specific capacitance of up to 433.02 μF·cm^−2^ at 5 μA·cm^−2^ was reported, and, simultaneously, the specific energy density reached a value of 1.73 × 10^−2^ μWh·cm^−2^. Moreover, the maximum achieved specific power density was 5.33 × 10^−1^ mW·cm^−2^ at 1 mA·cm^−2^ [122].

Recently, great advances in the impact of complex and intelligent machines are more noticeable, especially when it comes to the biomedical engineering sciences. Such is the case for computers, mobile devices, sensors, actuators, and robots. Because of this, unconventional materials have been designed for bridging between humans and machines through hydrogel interfaces in a broad range of applications [123].

### 5.5. Metallo-Supramolecular Hydrogels

One paper demonstrating ion size variation is that presented by Dubey and co-workers, who reported on a series of fluorescent metallohydrogels with attached alkali metal ions that show large conductance and rheological properties consistent with alkali metal ion size variation [124].

Following this order of ideas, the use of 3,6-bis(2-pyridyl)-1,2,4,5-Tetrazines (bptz) ligands for the synthesis of supramolecular gels based on metal–ligand coordination has been described, showing notable dynamic mechanical properties. These materials have the advantage of being easily functionalized via Diels–Alder reactions; therefore, metallohydrogel scaffolds have been prepared for the release of small molecules activated via photoactivation and enzymatically. The Diels–Alder functionalization of the bptz ligands attached to the ends of the PEG chains is ensured via the gelation induced by metal coordination in the presence of Ni^2+^ and Fe^2+^ cations [125].

One aspect of complex peptides is obtained via the coordination of the metal ions that impact their functionality; however, they are also participants in the self-assembly and, at the same time, in the structuring of supramolecular gels. In this context, His–copper (His-Cu) coordination in the supramolecular assembly of gelators is used to improve the understanding and development of these materials. Metal–gel coordination mimics biologically relevant metal–peptide coordination, influencing hydrogel self-assembly and their mechanical properties, biodegradability, biocompatibility, adjustability, and recycling, while metal coordination allows their widespread applications in the biomedical, waste management, and catalysis industries [126]. This has led to studies reporting a peptide-based simple drug delivery system (DDS) for doxorubicin (DOX) delivery as an anticancer system. This DDS is structured with tripeptides that form a metal–peptide coordination center with Cu(II) to self-assemble and thus generate structures whose morphology allows the controlled and sustained release of DOX from the His residue at the site [127]. The choice of His to displace drug molecules from the metallopeptide network is because, among the essential amino acids, His can easily bind strongly to the Cu^2+^ center. Regarding the above, for example, it is noteworthy that one of the four peripheral ZnII ions is anchored by the Glu and His side chains from one monomer and from another His [128]. Thus, the Lys amine group completes the tetrahedral coordination geometry. In another study, the kinetic and thermodynamic characteristics of pyridinedicarboxamide (PDCA) complexes with a series of transition metal ions were established, which could provide the basis for the development of new metallo-supramolecular polymeric networks and hydrogels. By means of the condensation of linear PEG segments with the PDCA ligand via urethane bonds and the complexation of PDCA with various metal ions under basic conditions, the rheological peculiarities were investigated [129].

The strength, stability, and connectivity of coordination bonds can be tuned largely without significant synthetic effort, solely through the simple choice of metal ion. Thus, it has been shown that energy dissipation modes are relevant in metallo-supramolecular systems, even more so by introducing a mixture of metal ions with different complex stabilities [130]. To a large extent, different binding constants for bis(terpyridine) and Mn(II) and Zn(II) complexes have been demonstrated. Multivalent effects are influenced by increasing the binding affinity [109]. Through connectivity mismatches, supramolecular hydrogels have been developed by systematically introducing different percentages of connectivity defects into a model supramolecular network, and the resulting macroscopic elastic response via oscillatory shear rheology, microscopic autodiffusivity via fluorescence recovery after photobleaching (FRAP), and characterizing and testing the specific type of defects via double-quantum NMR (DQ-NMR). In the work developed by Nicolella et al., it is highlighted that with connectivity defects, a hydrogel becomes softer and autodiffusivity increases inside [131].

In order to visualize the modes of energy dissipation, hydrogel models with metal–ligand coordination have been developed with divalent metal ions in the formation of a stable biscomplex, evidencing significant linearity in the energy dissipation. Deviation from linear behavior is to be expected in cases where metal ions and ligands show weak coordination affinity and even more so if metal ions compete to form mono, bis, or tris complexes [132].

Molecular interactions were observed when producing a metallogel with vanadium pentoxide. The metallogel was obtained by reacting (E)-N’-((2-hydroxynaphthalen-1-yl)methylene)benzohydrazide with vanadium pentoxide in a methanol solution. Surprisingly, a ligand gelation test showed that the ligand could not gel in both polar and nonpolar solvents. However, when the warm DMF solution of the ligand was reacted with the warm vanadium pentoxide solution in DMF/H_2_O (2:1 *v*/*v*), spontaneous gelation occurred when cooled to room temperature [133]. Metallogels show weak C–H---O and N–H---O bonding interactions. In a molecular packing diagram, it is possible to observe that the dioxovanadium (V) molecules are linked by N–H---O intermolecular hydrogen bonds to form a one-dimensional zig-zag linear arrangement.

In another case, amphiphilic peptides (APs) were studied for the construction of molecular blocks, especially for the manufacture of supramolecular soft materials with potential for various applications in the fields of science and technology. Especially in recent years, various amphiphilic peptides have been designed and synthesized [134], which is schematized in Figure 10. Amphiphilic peptides are divided into two different subclasses, namely, peptides containing alternating hydrophilic/hydrophobic amino acid residues and those containing a long hydrophobic stretch of amino acids attached to a hydrophilic sequence. Obviously, the impacts on the aggregation properties of these designer amphiphiles vary markedly.

Metallo-supramolecular polymer networks (MSPNs) are the subject of study in order to detail certain biomimetic functions such as self-healing and stimuli-responsiveness. A good example is the use of His, which is an α-amino acid of great importance in enzyme function. It is found abundantly in the soft, collagen-like cores of the byssus threads of marine mussels. It contains an a-amino group capable of being protonated under biological conditions and a carboxylic acid, in addition to containing an imidazole group. The nitrogen atom of the imidazole group is protonated at a pH of <6 (pKa 6.5), while under neutral and basic conditions, it falls apart, which is a magnificent property. Thus, depending on the pH conditions, His may have the feasibility of forming compounds with various transition metal ions and shaping various coordination geometries.

Metal ions are essential in controlling the structural properties and catalytic functions of many proteins required for proper physiological function. Thus, artificial metalloproteins are built by introducing metallocofactors into a natural protein. Detailed examples of the bioinorganic active sites of cupredoxins, commonly known as blue copper proteins, are presented by Borovik and his team [135]. The transfer of electrons between the Cu^2+^ and Cu^+^ redox states causes the active sites of cupredoxins to present a distorted trigonal monopyramidal coordination geometry in the Cu(I) and Cu(II) states. The highly covalent nature of the Cu−S_cys_ bond in Cys (cysteine) results in charge transfer from the Sπ−Cu ligand to the metal, which is responsible for the characteristic blue color (λ_max_ ≈ 600 nm) and the small hyperfine coupling constant observed using EPR spectroscopy (A ≈ 180 MHz).

## 6. Conclusions and Perspectives

We have summarized the main synthetic strategies and recent functional applications of supramolecular chemistry based on hydrogels and their derivatives, which are a big topic in the hierarchy of materials experiencing explosive growth, especially in the last two decades, and the situation continues. In view of the above, a wealth of peculiar information about supramolecular hydrogels visualizes the non-covalent interactions that modulate gel properties and establishes many opportunities for current and future research to make the most out of combining a larger number of supramolecular interactions to optimize many characteristics of hydrogels in order to design cautiously tailored properties. Moving forward, the fine collaboration between scientists should be visualized and emphasized at the same time as the higher-performing and realistic biomimetic materials. Without being onerous, the research on tissue engineering should additionally incorporate biocompatibility and should include in vivo animal studies to monitor the long-term immune response. In this way, ophthalmic issues and adhesiveness that have to do with biological task are addressed.

Regarding other ideas, this review reported on classic topics such as hydrogels that ranged from their relationship with drug delivery systems to metallo-supramolecular materials, through to their applications in ophthalmic systems and in applications that require adhesiveness and electrically conductive hydrogels. This review also addressed the hydrogel systems called TENGs with ionic conductive capacities, stretchability, flexibility, and biocompatibility, and, more generally, electrically conductive hydrogels were discussed throughout this review. Of course, the exciting topic of self-healing hydrogels was not left aside. The requirements that are relevant to self-healing hydrogels are phase mobility, the requisitioned rate of self-recovery, multiple healing cycles, and, usually, autonomous healing in different environmental conditions, such that a common hydrogel with a dynamic bond would not fulfil each aforesaid requirement.

With the expanding fundamental understanding of the capacity of hydrogels to improve our daily lives, hydrogels for drug delivery systems, self-healing hydrogels, and electrically conducting hydrogels are likely to further change the scale, efficacy, and cost of the constitutive effort in areas not considered within the academic and industrial fields.

## Figures and Tables

**Figure 1 polymers-15-01365-f001:**
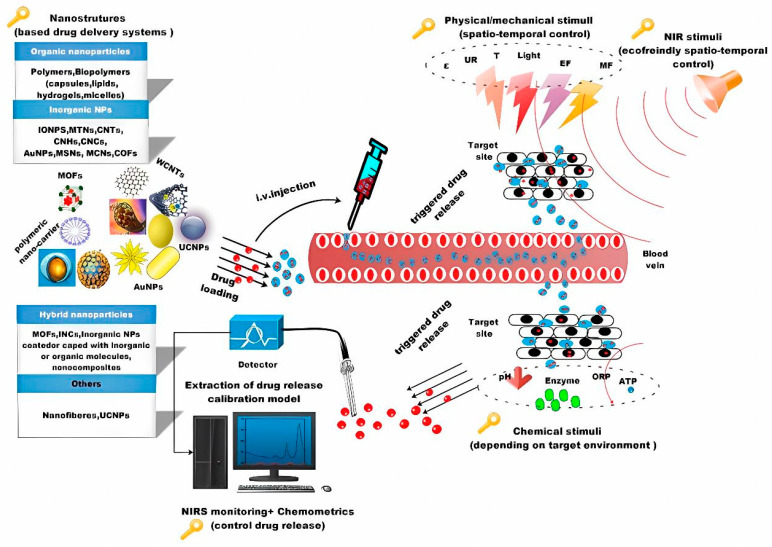
Basic nanostructures for DDSs and different stimuli. Reprinted with permission from Ref. [21].

**Figure 2 polymers-15-01365-f002:**
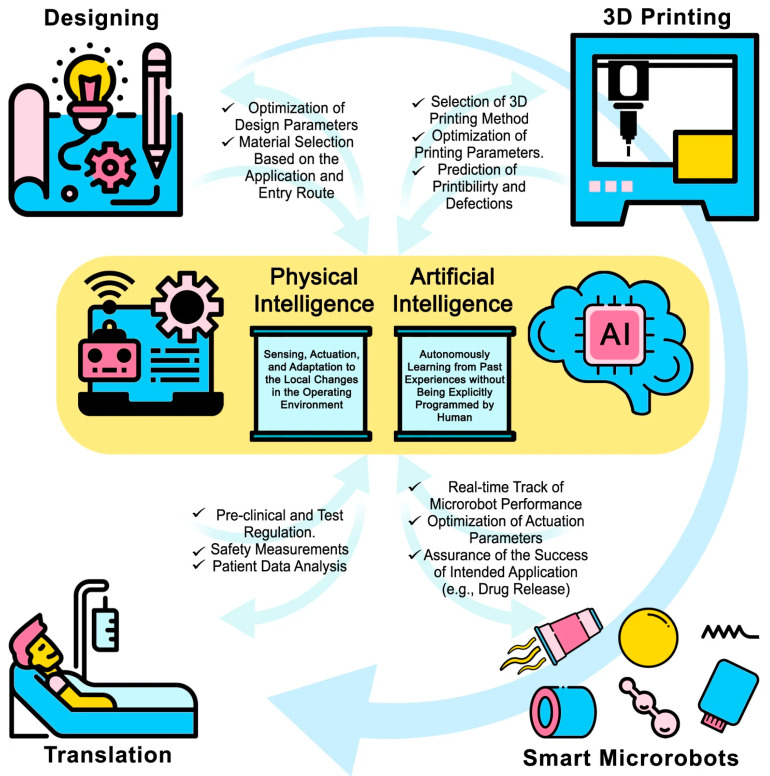
Proposed design of a microrobot. The material selection can be assisted by AI and PI to design a particular device. Artificial intelligence (AI) can help clinicians to track microrobots and enhance maneuverability by tuning actuation parameters to ensure proper functionality, while physical intelligence (PI) enables microrobots to sense different stimuli in their environment and respond to those stimuli independently, for instance, drug release at a particular pH level at the target site. In this regard, test procedures must be regulated to maintain security factors. Reprinted with permission from Ref. [54].

**Figure 3 polymers-15-01365-f003:**
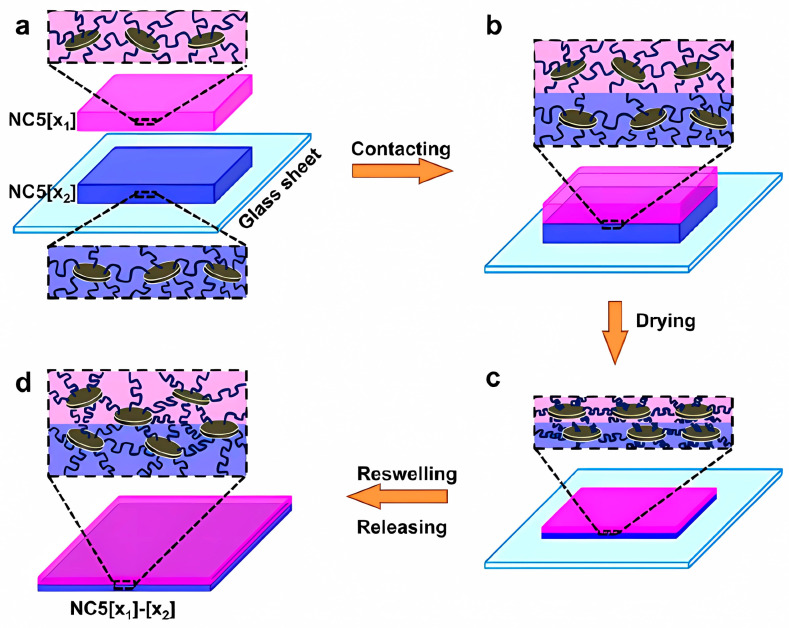
Schematic illustration of the assembly process of subunits for constructing the hydrogel NC5[x_1_]–[x_2_] via rearranged hydrogen bonding between polymers and clay nanosheets; poly(N-isopropylacrylamide-co-acrylamide) (poly-(NIPAM-co-AM)) hydrogel is coded as NC5[x]: NC represents nanocomposite, “5” represents the clay concentration in the hydrogel fixed at 5 × 10^−2^ mol L^−1^, and “x” stands for the molar percentage of acrylamide (AM) content in the total monomer (**a**). The nanocomposite subunits NC5[x_1_] and NC5[x_2_] hydrogels are stacked together on a glass plate (**b**). During the drying process under atmospheric condition of 70% relative humidity at room temperature, the hydrogen bonding between polymers and clay nanosheets of NC5[x_1_] and NC5[x_2_] hydrogel subunits at the interface are rearranged to thoroughly interlock the two subunits (**c**). NC5[x_1_]–[x_2_] hydrogel with inhomogeneous structure is formed due to the strong hydrogen bonding between polymers and clay nanosheets at the interface after reswelling in deionized water (**d**). Reprinted with permission from Ref. [61].

**Figure 4 polymers-15-01365-f004:**
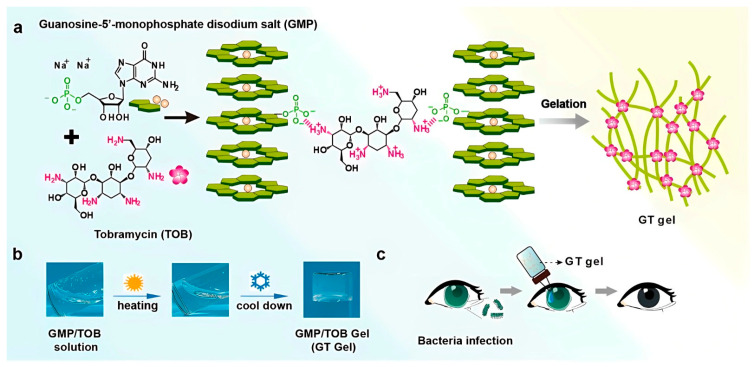
(**a**) Illustration of GMP-TOB Gel (GT Gel) formation via electrostatic interactions between GMP-based G-quadruplex and tobramycin. (**b**) GT Gel preparation scheme, the process involves heating and cooling. The final objective of the GT gel is the treatment of bacterial keratitis (**c**). Reprinted with permission from Ref. [74].

**Figure 5 polymers-15-01365-f005:**
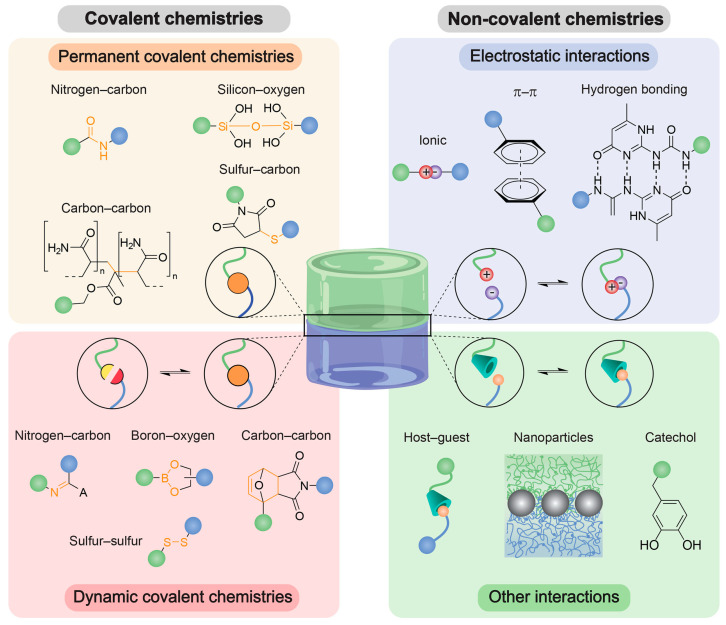
Overview of common bonds that are used as adhesion junctions. Adhesion junctions can be formed via permanent or dynamic covalent or noncovalent bond formation. Noncovalent interactions include electrostatic interactions and other interactions such as host–guest chemistries and polymer−nanoparticle (NP) interactions, for instance, catechol-based ones. Reprinted with permission from Ref. [7].

**Figure 6 polymers-15-01365-f006:**
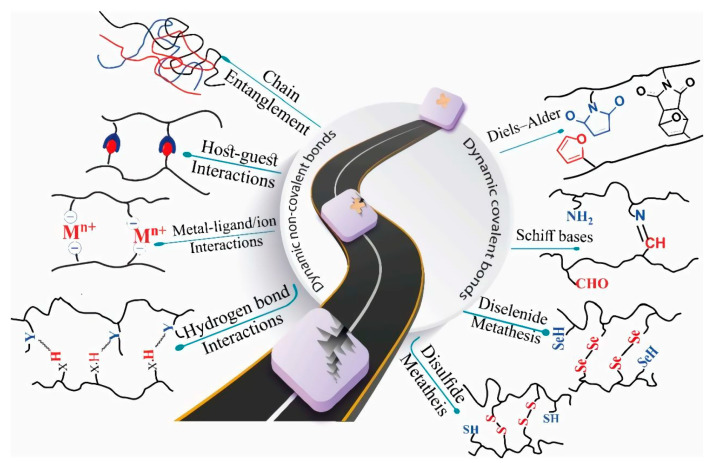
Schematic of dynamic covalent and no-covalent bond operation in the self-healing process. Reprinted with permission from Ref. [102].

**Figure 7 polymers-15-01365-f007:**
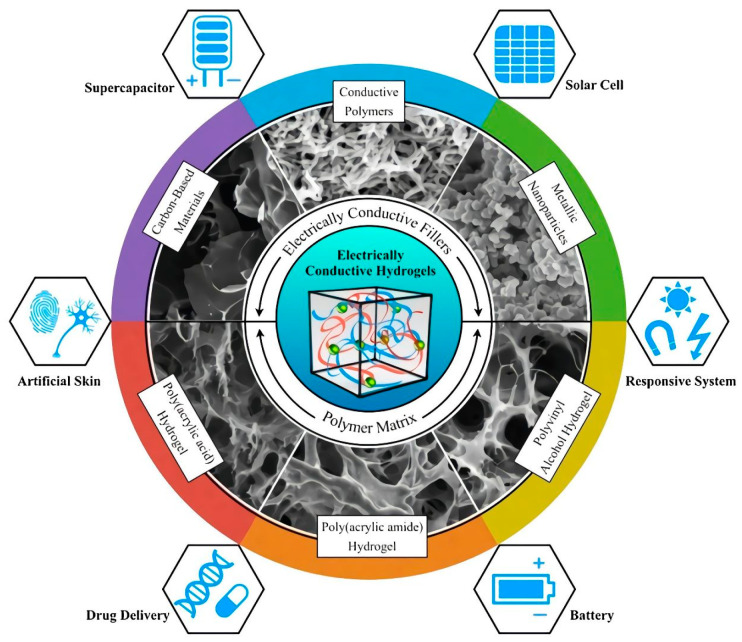
Electrically conductive hydrogels are an emerging class of hydrogels combining a hydrophilic matrix with conductive fillers, and they have exceptional promise in a wide range of applications. Reprinted with permission from Ref. [116].

**Figure 8 polymers-15-01365-f008:**
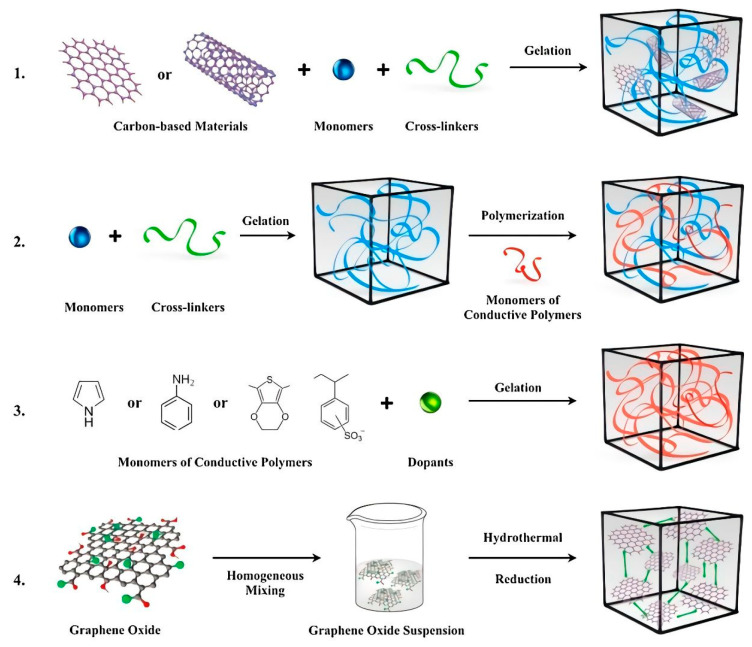
Four main approaches used to obtain ECAs: (**1**) hydrogel formation via a conductive filler suspension; (**2**) polymerization within a preformed hydrogel matrix; (**3**) cross-linking conductive polymers with dopant molecules; and (**4**) self-assembly of graphene hydrogel via supramolecular interactions. Reprinted with permission from Ref. [116].

**Figure 9 polymers-15-01365-f009:**
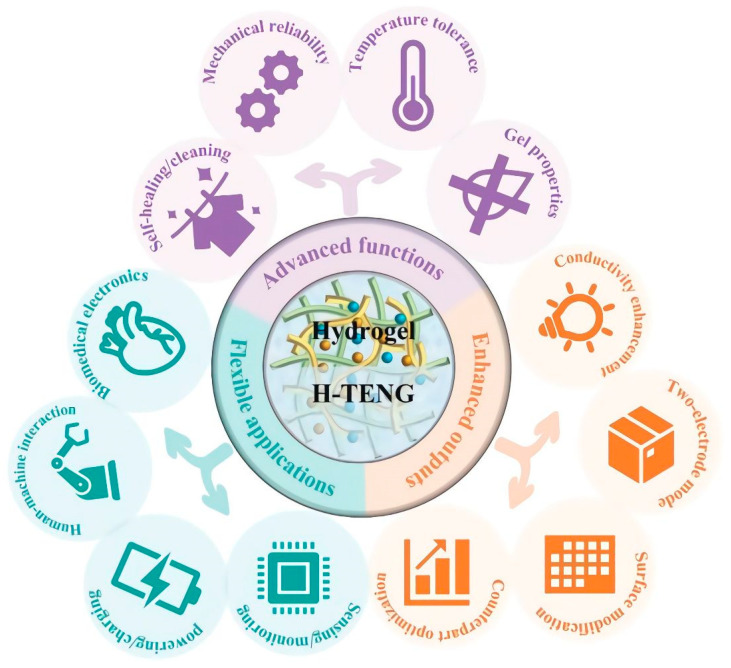
Development of H-TENGs: advanced functions, enhanced outputs, and flexible and wearable applications. Reprinted from Ref. [121] with permission.

**Figure 10 polymers-15-01365-f010:**
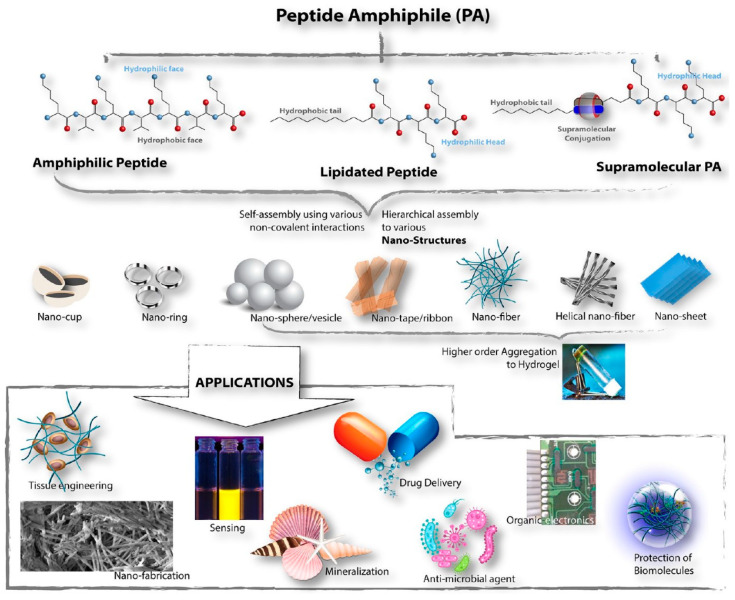
Schematic overview of the peptide amphiphile (PA) classification, different self-assembled nanostructures formed, and PA applications [134].

## Data Availability

Not applicable.

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
