# Peer review of "Structural Strategies for Supramolecular Hydrogels and Their Applications"

_polymers, 2023, doi:10.3390/polym15061365_

Round 1

Reviewer 1 Report

Author of this manuscript provided a comprehensive review on hydrogels refering 130 references. In reality, however, thousands of papers have been published on supramolecular hydrogels, including reviews. Nevertheless, the authors have found their niche and summarised the main synthetic strategies for hydrogels and their most prominent applications. The focus in this manuscript was on self-healing hydrogels, adhesive hydrogels, conductive hydrogels, clays and 3D structures.

The author discusses some ways of forming hydrogels, of which there are essentially two: non-covalent intermolecular dynamic bonds and covalent cross-linkinng and gives predominance to the numerous possibilities of practical application of hydrogels. Some of them are of special practical importance. Preparation of drug delivery systems based on hydrogels with controlled release properties leads to creation of enhanced biomaterials and contribute to the progressive creation of dot-hydrogels for drug delivery. Many polymers were studied in this direction, some of them were of natural origin, for instance, chitin, cellulose, alginates, with chemically and physically cross-linking  domains.

The second type of hydrogels described by the author is a way to make 3D printing of ionically cross-linked liquid crystalline hydrogels from water soluble polymers. A number of hydrogels have been prepared and analysed. One of the method to prepare this type of hydrogels is as follows, to make intermolecular interactions between two polymers, which formed the self-assembled structures before interaction. These hydrogels can be prepared with different methods and can be used in a number of application areas, especially, in medicine and construction of new self-healing hydrogel systems, capable to reform their dynamic, biochemical, and mechanical gel properties.

There is no need to look in detail at the other sections of the manuscript that are listed at the beginning of the review, as they are all described in the same way. The paper is professionally written, covers a wide range of issues, especially in terms of the practical application of hydrogels, is very informative and has no major shortcomings. The manuscript's subject matter is well in line with the direction of Polymer. It can, without a doubt, be submitted for publication in the journal.

The wide range of issues that the manuscript covers is, in a way, a disadvantage. I only want to express my personal opinion, namely, that it is inadvisable for a review of this kind to cover too many issues and deal with too many different types of polymers. In those reviews, it is difficult to work out one's own strategy for obtaining them. I find more useful for readers such reviews that focus either on certain classes of polymers (e.g. synthetic or natural) or on certain strategic lines for making them.

Nevertheless, I believe that this manuscript deserves to be published.

Author Response

Reply 1. Thank you for your positive comments and for taking the time to read the manuscript carefully. Without a doubt, your comments are an incentive to be distinctive in our investigations. In addition, your comments also have an impact on my work group.

Reviewer 2 Report

1. It is suggested that some of the lines in the document need to be revised for a better understanding of the author's view on literature references, as it should not be direct copy pasted from literature (since its a review article). For example - Rewrite the abstract: Line 17 to 20

Rewrite the sentences for better understanding and continuity :

Lines 37 to 39 etc.

Please review the complete manuscript.

2. THe article has focussed more on applications of  supramolecular hydrogels and their derivatives rather than structural strategies or how structure affects the function of hydrogel.

A small paragraph on mechanism structure-property-function and chemistry of hydrogels can be given in the introduction before describing them in detail. It is recommended that please put the author's view in this paragraph, after understanding from literature.

3. Index at the beginning of the chapter would be helpful to follow and understand the idea of the manuscript

4. SInce hydrogels can be made using different chemistries/combinations how are supramolecular hydrogels different from other types? either structurally or functionally? 

Author Response

Comment 1. It is suggested that some of the lines in the document need to be revised for a better understanding of the author's view on literature references, as it should not be direct copy pasted from literature (since its a review article). For example - Rewrite the abstract: Line 17 to 20

Rewrite the sentences for better understanding and continuity:

Lines 37 to 39 etc.

Please review the complete manuscript.

Reply 1. I thank you for taking the time to read the manuscript in detail. I have responded to your suggestions; you can see the changes in the abstract and in lines 38-42. In addition to improving other paragraphs with dubious writing.

Comment 2. The article has focused more on applications of supramolecular hydrogels and their derivatives rather than structural strategies or how structure affects the function of hydrogel.

A small paragraph on mechanism structure-property-function and chemistry of hydrogels can be given in the introduction before describing them in detail. It is recommended that please put the author's view in this paragraph, after understanding from literature.

Reply 2. In this sense, in response to your invaluable suggestion, I have made substantive changes that you can see in lines 56 to 69.

Comment 3. Index at the beginning of the chapter would be helpful to follow and understand the idea of the manuscript.

Reply 3. Following your assertive suggestion, I have made the paragraph with a text that you will see on lines 111 to 119, hoping that it will be to your liking.

Comment 4. Since hydrogels can be made using different chemistries/combinations how are supramolecular hydrogels different from other types? either structurally or functionally?

Reply 4. Following his assertive comment that I have attended, together with his proposal 2, with an additional paragraph in the manuscript (lines 56-69), of course, hoping that it is to his liking.

Reviewer 3 Report

The review article entitled “Structural strategies of supramolecular hydrogels and their applications” features a systematic review of methods for the supramolecular structures are of great interest due to their applicability in various scientific and industrial fields. This review also addresses systems that are largely based on hydrogel chemistry and the enormous opportunities to design very specific structures for applications that demand enormous specificity. Thus, issues are needed to be addressed first before the recommendation of this review article for publication to Polymer.

Comments

1.    The author should mention the year of literature collections in the abstract section.

2.    Introduction section need to be improved with suitable references.

3.    The author should provide the suitable schematic representation/figure for the following sections; 3. D-printed hydrogels, 4. Clays into hydrogels, 5.1. Ophthalmic hydrogels, 5.2. Adhesive hydrogels, 5.5. Metallo-supramolecular hydrogels.

4.    The manuscript contains some typographical errors and superfluous spaces that need to be corrected accordingly.

Author Response

The review article entitled “Structural strategies of supramolecular hydrogels and their applications” features a systematic review of methods for the supramolecular structures are of great interest due to their applicability in various scientific and industrial fields. This review also addresses systems that are largely based on hydrogel chemistry and the enormous opportunities to design very specific structures for applications that demand enormous specificity. Thus, issues are needed to be addressed first before the recommendation of this review article for publication to Polymer.

Comment 1. The author should mention the year of literature collections in the abstract section.

Reply 1. I appreciated you taking the time to thoroughly read the manuscript. In response to your assertive suggestion, I have putted that references regarding supramolecular hydrogels can be obtained from Web of Science which you can see in the abstract, please you see the lines 18 to 23. 

Comment 2. Introduction section need to be improved with suitable references.

Reply 2. Following his assertive comment, I have attended to your proposal. In the text you can see the references [7], [13], [14], [15], and [16] that I have put in the introduction additionally.

Comment 3. The author should provide the suitable schematic representation/figure for the following sections: 3. D-printed hydrogels, 4. Clays into hydrogels, 5.1. Ophthalmic hydrogels, 5.2. Adhesive hydrogels, 5.5. Metallo-supramolecular hydrogels.

Reply 3. I appreciate your suggestion; it will undoubtedly enhance the quality of the manuscript. In this regard I have added Figures 3, 4, 5 and 10.

Comment 4. The manuscript contains some typographical errors and superfluous spaces that need to be corrected accordingly.

Reply 4. Thank you very much for the comment, in this regard I have revised the manuscript in order to correct those errors, as you know at the time of writing, sometimes it is possible to make mistakes and not see them, until someone, like you, perceives them.